Splicing Express: a software suite for alternative splicing analysis using next-generation sequencing data

Kroll Jose E. 1 2
Kim Jihoon 3
Ohno-Machado Lucila 3
de Souza Sandro J. 2 sandro@sandrodesouza.com.br
1 Institute of Bioinformatics and Biotechnology , Natal, Rio Grande do Norte , Brazil
2 Brain Institute, Universidade Federal do Rio Grande do Norte , Natal, Rio Grande do Norte , Brazil
3 Department of Biomedical Informatics, University of California, San Diego , La Jolla, CA , United States
Goyal Pankaj
Electronic publication date: 2015 Nov 19
Publication date: 2015
Volume: 3
Electronic Location ID: e1419
Received 2015 Jul 31; Accepted 2015 Oct 28
Copyright: © 2015 Kroll et al.
Copyright year: 2015
Copyright holder: Kroll et al.
License: This is an open access article distributed under the terms of the Creative Commons Attribution License, which permits unrestricted use, distribution, reproduction and adaptation in any medium and for any purpose provided that it is properly attributed. For attribution, the original author(s), title, publication source (PeerJ) and either DOI or URL of the article must be cited.
License URL: https://creativecommons.org/licenses/by/4.0/

Keywords: Transcriptome, Bioinformatics, RNA-Seq, Next-generation sequencing, Alternative splicing, Visualization, Suite tool, Software, Expression

Funding: CNPq 501891/2013-7 483775/2012-6 CAPES edital 051/2013 NIH D43TW007015 U54HL108460 JEK is supported by a post-doctoral fellowship from CNPq (501891/2013-7). This research was supported by grants from CNPq (483775/2012-6) and CAPES (edital 051/2013), both to SJS, and grants D43TW007015 and U54HL108460 from NIH to LOM. The funders had no role in study design, data collection and analysis, decision to publish, or preparation of the manuscript.

==============================
Motivation. Alternative splicing events (ASEs) are prevalent in the transcriptome of eukaryotic species and are known to influence many biological phenomena. The identification and quantification of these events are crucial for a better understanding of biological processes. Next-generation DNA sequencing technologies have allowed deep characterization of transcriptomes and made it possible to address these issues. ASEs analysis, however, represents a challenging task especially when many different samples need to be compared. Some popular tools for the analysis of ASEs are known to report thousands of events without annotations and/or graphical representations. A new tool for the identification and visualization of ASEs is here described, which can be used by biologists without a solid bioinformatics background.

Results. A software suite named Splicing Express was created to perform ASEs analysis from transcriptome sequencing data derived from next-generation DNA sequencing platforms. Its major goal is to serve the needs of biomedical researchers who do not have bioinformatics skills. Splicing Express performs automatic annotation of transcriptome data (GTF files) using gene coordinates available from the UCSC genome browser and allows the analysis of data from all available species. The identification of ASEs is done by a known algorithm previously implemented in another tool named Splooce. As a final result, Splicing Express creates a set of HTML files composed of graphics and tables designed to describe the expression profile of ASEs among all analyzed samples. By using RNA-Seq data from the Illumina Human Body Map and the Rat Body Map, we show that Splicing Express is able to perform all tasks in a straightforward way, identifying well-known specific events.

Availability and Implementation.Splicing Express is written in Perl and is suitable to run only in UNIX-like systems. More details can be found at: http://www.bioinformatics-brazil.org/splicingexpress.

Introduction

Alternative splicing is involved in many biological phenomena and has been better evidenced due to the development of next-generation sequencing (NGS) technologies (Trapnell, Pachter & Salzberg, 2009). RNA-Seq technologies have facilitated the deep characterization of transcriptomes of several species. However, the huge amount of data generated in each experiment brings challenges for the analysis and interpretation of alternative splicing. This is particularly critical for researchers or research groups that lack bioinformatics expertise. Despite the fact that some bioinformatics tools have already been developed for the identification of ASEs in NGS-derived data (Liu et al., 2012; Seok et al., 2012; Florea, Song & Salzberg, 2013), challenges remain. For example, most of the tools available rely on complex interfaces or command line inputs, not to mention their complex output. Overall, there is a lack of user-friendly tools that are suitable for genome-wide ASE analysis, especially when several samples need to be compared.

Here we present a software suite named Splicing Express, which was designed to provide a simple and effective way for the identification, annotation and visualization of ASEs. It requires GTF files as input, as those created from Cufflinks (Roberts et al., 2011). By using RNA-Seq data from the Illumina Human Body Map Project and the Rat Body Map (Yu et al., 2014), we show that Splicing Express is able to extract meaningful information from deep transcriptome data. Both analyses using Splicing Express are available at http://www.bioinformatics-brazil.org/ASE_HBM.

Material and Methods

Sequences

The human genome reference sequence (GRCh37/hg19) was downloaded from UCSC Genome Bioinformatics portal (http://genome.ucsc.edu), as well as RefSeq sequences and their respective annotations (files refFlat.txt.gz and refSeqAli.txt.gz). RNA-Seq data from the Illumina Human Body Map Project Version 2 were downloaded from the EMBL-EBI portal (accession E-MTAB-513). All available tissues were used: adrenal, adipose, brain, breast, colon, heart, kidney, liver, lung, lymph, ovary, prostate, skeletal muscle, testes, thyroid and white blood cells. Analyses were restricted only to the 2 × 50 bp paired-end sequences, which represent approximately 80 million read pairs per tissue.

Transcriptome assembly

Reads from all 16 tissues from the Illumina Human Body Map Project were mapped to the genome using Tophat version 2 (Trapnell, Pachter & Salzberg, 2009). A conservative parameter was maintained (–splice-mismatches 0), and a reference GTF file, containing RefSeq aligments from the UCSC portal, was set for helping the mapping process. The resulting BAM files created by Tophat were, afterwards, used as input for Cufflinks version 2 (Roberts et al., 2011). Use of that software was carried out using default parameters and without any reference file as input. Additional processing, such as sequence clustering, was performed by Splicing Express.

Sequencing clustering

The expressed sequences were clustered by Splicing Express, using a strategy that has been used previously by us (Galante et al., 2004; Galante et al., 2007; Kroll et al., 2012). Basically, the genome coordinates of all expressed sequences were compared to the genome coordinates of Reference Sequences (RefSeqs) entries. Sequences presenting multiple exons and sharing at least one exon-intron boundary with a RefSeq sequence were merged. In the specific case of Splicing Express, all remaining sequences were excluded from the alternative splicing analysis, but were taken into account for measuring the total gene expression. The intronless sequences were merged by checking their overlap to RefSeqs, which should be greater than or equal to 30 bp.

Alternative splicing analysis

The GTF files resulting from the transcriptome assembly of the Illumina Human Body Map Project were processed using Splicing Express. The expressed sequences were clustered to a set of RefSeq sequences, which comprehended 26.318 genes and a total of 50.129 entries. The processing steps are better described in the “Implementation.” In-house Perl scripts were developed for all large-scale analyses, and all events described in the “Analysis” were visually identified by using Splicing Express. A basic analysis using Splicing Express was also performed for the Rat Body Map (Yu et al., 2014), which is available at the Splicing Express website.

Statistics

Splicing Express performs statistical analysis aiming to give significance values to the fold differences between the splicing variants. That analysis is only performed when the GTF files have data regarding the FPKM confidence intervals, which are standardly present in any GTF file created by Cufflinks (Roberts et al., 2011). Those values are gathered to calculate the FPKM Standard Error (SE), given by the equation: SE=Chi−Clo/2∗1.96,

where Chi and Clo are the FPKM values at the lower and higher limit of the 95% FPKM confidence interval, respectively. Moreover, it is known that more than one expressed sequence can share the same splicing event in a sample. Hence, the FPKM of those expressed sequences have to be summed. Its respective SE is then calculated by the equation: SEx=SE12+⋯+SEn2,

where x refers to a splicing cluster and n refers to the expressed sequences from the respective cluster x. Finally, a Welch’s t-test is performed between the two clusters of splicing variants, defined here as a and b. The t-test equation is given by: t=Xa−Xb/SEa2+SEb2,

where X is the FPKM of the sample. Probabilities are then calculated from t-values through the Perl Statistics::Distributions library using a degree of freedom equal one.

Availability

Splicing Express source code is available through a repository manager platform at http://www.bioinformatics-brazil.org/splicingexpress, where tutorials (written and video), sample files, and a bug track system are provided.

Results and Discussion

Implementation

Splicing Express is a suite composed of Perl scripts that executes several processing steps, as schematized in Fig. 1B. Prior to running Splicing Express, parameters about the reference genome, the output directory and the input GTF files must be firstly set. After that, GTF files are checked for consistency and converted to a workable file format. GTF files created by Cufflinks are recommended to be used as input due to some annotation standards referring to sequence names and FPKM expression. Similarly, chromosome names and genome positions from the GTF files must be compatible with the respective genome data provided by the UCSC genome portal.

Figure 1 Overview of Splicing Express features.

(A) Table generated by Splicing Express showing expression level and number of variants in a gene-by-gene basis. (B) Schematic representation of the computational pipeline used by Splicing Express. (C) Graphic represention of the expression pattern of NRXN3. (D) Graphic represention of the differential expression of one isoform (exon skipping) of NRXN3. The exon skipping isoform is represented in red, while the isoform including the respective exon is colored in green.

Expressed sequences are then clustered through a known strategy (see “Material & Methods”). Clustering is needed for identifying the gene of origin of each expressed sequence contained in the GTF files, and is also very important to standardize the annotation process using gene symbols according to the HUGO (Gene Nomenclature Committee). Splicing Express, therefore, allows the use of GTF files showing any kind of gene annotation since those information will be posteriorly rewritten.

The ASEs identification is done through an algorithm based on pairwise comparison, a very common approach used for many other similar tools (Florea, Song & Salzberg, 2013; Kroll et al., 2012; Sammeth, Foissac & Guigó, 2008). Splicing Express, moreover, uses an strategy implemented previously by us in Splooce (Kroll et al., 2012), which is based on regular expressions that are capable of identifying all well-known simple and combined ASEs. In that strategy, the expressed sequences are represented as binary sequences, where exons and introns are represented by 1 and 0, respectively. Those sequences are then pairwisely compared in an exhaustive manner, creating numerical patterns which represent their splicing differences. Currently, Splicing Express is able to identify patterns of simple alternative splicing events, such as exon skipping, intron retention and alternative 5′ and 3′ splicing borders.

Afterwards, a graphic representation for the expression level (FPKM) is created for each gene and for each identified ASE. Graphics are created using an external SVG library, which is available at CPAN (Comprehensive Perl Archive Network). They can be opened in all recent releases of internet browsers, zoomed in or out without loss of quality. Finally, those graphics are gathered together with additional information in HTML files. An example of some graphical displays are shown in Figs. 1C and 1D. A main file (index.html) is created at the root of the results directory tree, providing advanced tools for exploring and filtering events (Fig. 1A). A basic statistical summary and a help section are also provided.

Performance

Splicing Express analysis does not aim speed or efficiency since the major computational processing is done previously by tools like Tophat and Cufflinks. Overall, Splicing Express processing time is fast, and it is able to process 15 samples in about half-hour using an average desktop personal computer. Here, we performed a simple performance analysis. No accuracy and sensitivity parameters were measured because the ASE identification algorithm here proposed is based on an exhaustive approach. As a testing platform, an i7 Core processor with 16 GB RAM machine was chosen, although Splicing Express uses only a single core and needs no more than 1 Gb of RAM memory.

Sixteen GTF files representing each tissue from the Illumina Human Body Map Project were analyzed. The performance was measured for each processing step of Splicing Express as a function of the number of analyzed samples (Fig. 2). The processing was done in triplicate and the results shown represent their average. Total processing time increased almost linearly as more samples were added to the analysis. The linear trend is particularly shaped by the processing time of “Data preparation” and “Annotation” steps. The “ASE identification” step, on the other hand, is based on a pairwise strategy, which may consume exponentialy more processing time as more samples are analyzed. The “ASE identification” processing, however, was very fast and the exponential trend could be barely observed. In turn, the “HTML & Graphics” processing time theoretically decreases as the identification of new alternative splicing events becomes saturated (less graphics are created), meaning that it tends to stabilize as more samples are analyzed. However, it could not be observed for the current data. “HTML & Graphics” step also depends on a moderate use of disk I/O due to a large amount of files being written. Hence, slow downs may be observed when outputing Splicing Express results directly to pen drives or other kind of slow devices.

Figure 2 Performance analysis for each processing step of Splicing Express as a function of the number of analyzed samples.

Analysis

Data from the Illumina Human Body Map Project was used for validating the use of Splicing Express in finding specific events among different samples. In total, 16 distinct tissues, showing high sequence coverage, were processed. As a result, 43,307 events were identified, from which 36% comprehended well-known events (Table 1). Meanwhile, partially known and unknown events accounted for 64% of the data, as expected (Florea, Song & Salzberg, 2013). Known events were defined as any event showing both variants supported by a RefSeq. Partially known events, in turn, were defined as events showing only one of both variants supported by a RefSeq. In order of frequency, taking into account only known events, the most frequent event type was exon skipping, followed by alternative splicing borders and intron retention, as observed before (Kroll et al., 2012). A more detailed analysis, considering tissue-specific alternative splicing events (tsASEs), showed that brain, adrenal gland, lymphnode and testis, among all analyzed tissues, had the highest amount of specific variants (t(df = 14) = 9.644117, p < 0.00001) (Table 2). TsASEs were defined as any splicing variant showing a relative expression greater than or equal to 75% of the total expression in all analyzed tissues. Only TsASEs showing a FPKM higher or equal than 1 were selected.

Table 1 Frequency of alternative splicing events by type and by degree of knowledge.

	Known	Partially known	Unknown	Total	
Exon skipping	9,175	8,147	1,012	18,334	
Alt. 5′ border	2,368	2,766	1,101	6,235	
Alt. 3′ border	2,755	3,845	1,019	7,619	
Intron retention	891	8,168	2,060	11,119	
Total	15,189	22,926	5,192	43,307	

To further explore the use of Splicing Express, we focused on the analysis of brain, since it is known to show one of the highest levels of skipped exons, alternative 5′ and 3′ splice sites (Yeo et al., 2004). Moreover, a high frequency of tsASEs was found for brain (Table 2). A gene ontology (GO) enrichment analysis of biological processes using DAVID (Huang, Sherman & Lempicki, 2009) showed that genes showing tsASEs for brain are more enriched with terms related to brain functions, such as “axonogenesis,” “cell morphogenesis involved in neuron differentiation” and “neuron projection morphogenesis” (p-value <6 × 10−3) (Fig. 3). MACF1, for example, is a microtubule actin crosslinking factor known to determine neuronal positioning by regulating microtubule dynamics and mediating GSK-3 signaling during brain development (Ka et al., 2014), and it was observed to be more expressed in 6 other tissues than brain. However, three ASEs were specific to brain (two exon skipping and one 3′ alternative splicing site). In genomic scale analysis, examples like that could only be related to a specific tissue through meticulous alternative splicing analysis, as that provided by Splicing Express.

Figure 3 Gene ontology (GO) enrichment analysis of biological processes for the 78 genes showing tsASEs in brain.

Table 2 Number of genes showing tissue-specific alternative splicing events (tsASEs).

Tissue	Exon skipping	Alt 3′ splice site	Alt 5′ splice site	Total	Unique	
Adipose	8	13	7	27	15	
Adrenal gland	21	31	32	80	39	
Brain	27	24	34	78	46	
Breast	11	12	13	32	13	
Colon	8	7	14	22	11	
Heart	13	14	13	40	17	
Kidney	11	16	21	44	24	
Leukocyte	11	18	18	41	16	
Liver	8	12	10	29	17	
Lung	10	22	14	46	26	
Lymphnode	22	39	26	87	49	
Ovary	13	15	17	40	22	
Prostate	16	14	13	40	21	
Skeletal muscle	16	10	6	28	15	
Testis	34	29	20	81	47	
Thyroid	20	21	17	51	33	

Among other events, an interesting one was observed for NRXN3, a neurexin related to presynaptic cell adhesion. In brain, the skipping/inclusion of exon SS#4 is known to control the postsynaptic AMPA receptor trafficking, being therefore an important factor for synaptic plasticity (Aoto et al., 2013). The specificity of NRXN3 and of its event to brain could be easily observed (Figs. 1C and 1D). On the other hand, many genes do not show tsASEs, such as the CELF family of proteins, which are involved in cell-specific and developmentally regulated alternative splicing in brain (Ladd, Charlet-B & Cooper, 2001). Splicing Express, however, was able to confirm the specificity of CELF3 and CELF5 to brain through their respective gene expression. For more informations about those and other genes, a full analysis of the Human Body Map and Rat Body map (Yu et al., 2014) using Splicing Express were made available at http://www.bioinformatics-brazil.org/ASE_HBM.

Conclusion

Splicing Express provides expression graphics in an intuitive fashion. The final HTML results can be open locally (offline), hence they can easily be hosted on web servers, shared and distributed. The complete automation in Splicing Express makes it a useful tool for deep analysis of ASE in any transcriptome. Now that deep sequencing is readily available and there is a shortage of human resources with bioinformatics skills, Splicing Express can make a significant contribution to the community.

We would like to thank Marbella Maria da Fonsêca for helping us on making the video tutorial, available at the Splicing Express website.

Abbreviations & Acronyms

ASE Alternative Splicing Event

DAVID The Database for Annotation, Visualization and Integrated Discovery

DNA Deoxyribonucleic acid

FPKM Fragments Per Kilobase Of Exon Per Million Fragments Mapped

HTML HyperText Markup Language

NGS Next-Generation Sequencing

RefSeq Reference Sequence

RNA-seq Whole Transcriptome Shotgun Sequencing

UCSC University of California, Santa Cruz

Additional Information and Declarations

Competing Interests

Author Contributions

Data Availability

The authors declare there are no competing interests.

Jose E. Kroll conceived and designed the experiments, performed the experiments, analyzed the data, contributed reagents/materials/analysis tools, wrote the paper, prepared figures and/or tables, reviewed drafts of the paper.

Jihoon Kim conceived and designed the experiments, contributed reagents/materials/analysis tools, reviewed drafts of the paper.

Lucila Ohno-Machado reviewed drafts of the paper.

Sandro J. de Souza wrote the paper, reviewed drafts of the paper.

The following information was supplied regarding data availability:

Code repository: http://www.bioinformatics-brazil.org/splicingexpress.

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
