# Peer review of "Splicing Express: a software suite for alternative splicing analysis using next-generation sequencing data"

_PeerJ, doi:10.7717/peerj.1419_

## Round 0.1 · original submission · Major Revisions

the reviewers have identified a number of issues that must be addressed before the paper can be re-considered for publication in PeerJ. The accuracy and reproducibility of the results should be presented and discussed thoroughly.

·

Basic reporting

Authors have presented a tool for analysis of alternative splicing events and named it as Splicing Express. I found this manuscript interesting and written in good style.

Experimental design

Authors have designed experiments in appropriate way. However two points can enhance manuscript quality:-

1) It is good to have a supplementary file describing how to use the Splicing Express with help of figures.
2) Authors have used human samples, it will add flavor to this manuscript if they use second organism as an example like zebrafish. This help more readers who work on other species than human.

Validity of the findings

Adhere to journal policies

Additional comments

I recommend some changes before it can be considered for publication with two points raised above.

·

Basic reporting

No Comments.

Experimental design

Please see the comments under "Validity of the Findings".

Validity of the findings

Splicing Express is a software that uses Splooce, another tool previously developed by the authors, to identify ASEs and creates HTML files of graphics and tables to describe the expression profile of the ASEs among all analyzed samples.

While this reviewer agrees with the authors that it is valuable to develop tools for the identification, annotation and visualization of ASEs, especially for biologists without bioinformatics background, I question the validity of the information presented by the current version of Splicing Express.

1. The accuracy and reproducibility of the results should be presented and discussed. The calculated expression levels of the isoforms and the fold changes shown on the website (http://www.bioinformatics-brazil.org/ASE_HBM/) vary tremendously. For example, the calculated expression levels of the two isoforms of NRXN3 shown in Figure 1D are 0.000 and 0.021 FPKM respectively in leukocytes, and their fold difference is shown as -0.009. Software such as Cufflinks would not have the accuracy and reproducibility for this situation. But apparently no information is given in the manuscript or on the website about the confidence of these values or the significance of the fold changes. A user would not be able to know how to interpret the information.

2. Similarly, what statistical criteria or cutoffs were used to identify the ASEs shown in Table 1 and 2?

3. The software currently shows the patterns of simple alternative splicing events, e.g. exon skipping, in samples, but more than one isoforms can share the same exon skipping event. Can the authors comment on how this is handled in the software?

---

## Round 0.2 · accepted · Accept

The manuscript has been improved significantly and meet the standard of PeerJ for publication.

·

Basic reporting

Standard and Good

Experimental design

Standard and Good

Validity of the findings

Standard - robust and sound

Additional comments

Given all points are addressed. I have no points to raise and good luck with this valuable work.